# Scaling models of visual working memory to natural images

Christopher J. Bates [1✉], George A. Alvarez[1] & Samuel J. Gershman [1]

Over the last few decades, psychologists have developed precise quantitative models of human recall performance in visual working memory (VWM) tasks. However, these models are tailored to a particular class of artificial stimulus displays and simple feature reports from participants (e.g., the color or orientation of a simple object). Our work has two aims. The first is to build models that explain people's memory errors in continuous report tasks with natural images. Here, we use image generation algorithms to generate continuously varying response alternatives that differ from the stimulus image in natural and complex ways, in order to capture the richness of people's stored representations. The second aim is to determine whether models that do a good job of explaining memory errors with natural images also explain errors in the more heavily studied domain of artificial displays with simple items. We find that: (i) features taken from state-of-the-art deep encoders predict trial-level difficulty in natural images better than several reasonable baselines; and (ii) the same visual encoders can reproduce set-size effects and response bias curves in the artificial stimulus domains of orientation and color. Moving forward, our approach offers a scalable way to build a more generalized understanding of VWM representations by combining recent advances in both AI and cognitive modeling.

[1] Department of Psychology, Harvard University, William James Hall, 33 Kirkland Street, Cambridge, MA 02138, USA. ✉email: cjbates@g.harvard.edu

When viewing an image, what details do we store in memory over the short term? What is the nature of the cognitive bottleneck that restricts how much information we can retain and recall? These and related questions have been pursued for the last several decades, leading to the discovery of several striking behavioral phenomena, including set-size, attraction, repulsion, and inter-item interaction effects[1,2]. Mathematical models offer compelling and principled explanations for many of these phenomena[3–8]. However, while these models can test competing theories about the nature of people's memory representations and capacity limits, they lack generality. Critically, they cannot predict what people will recall about *natural images*. While challenging, it is crucial to study memory for more ecological stimuli, since findings are likely to reveal important cognitive design principles that cannot be discovered by studying more simplified and artificial settings alone[9]. Moreover, given that our visual systems are optimized primarily to operate on natural images, it is reasonable to ask whether many of the phenomena we have identified in artificial domains are related to this adaptation.

The effort to study visual memory in more ecological settings is hindered in part by the same kinds of technological challenges facing much of vision science. Due to the visual system's complexity, we have long lacked precise models of the computations carried out in the visual stream. A simultaneous challenge lies in stimulus design. To probe the richness of our representations in the domain of natural images, we need methods to continuously vary stimuli in ways that appear natural to participants. Deep learning is beginning to offer effective tools to solve both of these problems.

To build a general computational account of VWM for natural images, we need a theory of where visual features come from. We argue that the most parsimonious hypothesis is that VWM is primarily built on top of feature detectors residing in the visual stream and that our memory systems efficiently reuse these computations by selecting subsets of the features and storing noisy or compressed versions of them. Arguably, the most precise models to-date for computations carried out along our visual streams come from certain classes of deep neural networks (DNNs)[10,11]. Thus, a reasonable starting place would be to select features from these networks as candidates for the features that feed into VWM.

In order to predict behavior, we next need to combine the selected deep neural network features with a noise model. Here, we adopt the target confusability competition (TCC) model[7]. This model is a generalization of a standard signal detection model, which assumes two response options, to tasks with arbitrary numbers of choices. Critical to our purposes, it can generate predictions for any feature space, including the kinds of complex, high-dimensional feature vectors that are likely needed to capture human visual representations, such as those derived from DNNs. The TCC model is flexible in this way because it relies only on pairwise similarity scores between the target stimulus and each response alternative. Thus, the stimuli can be represented in any hypothesized feature space, as long as a valid similarity metric can be applied. Incorrect responses are assumed to result from a noise process that corrupts the similarity scores (specifically, additive Gaussian noise).

We note that an alternative to TCC would be to add noise directly to the DNN representations, rather than to pairwise similarity scores. For instance, one could add Gaussian noise to each dimension of each DNN representation, then compute similarity scores using the noise-corrupted vectors, and finally take the maximum score as the response. This would lead to a model that is mathematically similar, but raises the complication that the model's behavior then depends on nuisance factors,

such as the dimensionality of the visual representations and statistical moments of the activation values. Here, we are most interested in whether the *representational geometry* of people's VWM representations is similar to that of a candidate DNN layer[12]. That is, does higher pairwise similarity in the DNN layer's representational space predict higher confusability in people's memories?

Our TCC-based models build on the original work in important ways. First, while Schurgin et al.[7] refit the model's single noise parameter ($d'$) for each set-size, here we show that feature spaces from select DNN layers can reproduce set-size effects without fitting separate noise parameters. Second, Schurgin et al. derived a psychological similarity function from perceptual similarity judgments, without identifying the origin of this similarity function. We show how DNNs can be used to derive similarity functions that are predictive of VWM for natural images. This also yields a practical benefit by obviating the need to collect pairwise similarity judgments, which is impractical for very large stimulus spaces.

We apply our modeling framework to VWM for both natural images and artificial stimuli (color and orientation), comparing several different DNN-based feature representations. To evaluate the models, we used a combination of quantitative metrics (correlation, likelihood) and qualitative checks (summary statistics derived from the models and data). We show that our framework can capture important aspects of both natural and artificial stimuli. However, the match between our models and human data does not yet approach the noise ceiling, suggesting room for improvement in future work.

## Methods

**DNN layers**. For ResNet-based models, we selected all layers from the pre-residual-block portion of the network, the last convolutional layer within each bottleneck sub-module in each residual block and the pooling layer just before the final output. For VGG-19, we selected the last 32 layers within the sub-module labeled "features" in the Torchvision implementation. This excluded the last two fully connected layers before the soft-max output. For the ConvNext models, we took the output of each `ConvNextLayer` within each `ConvNextStage`, as defined by the PyTorch model. For Vision Transformer-based models, we took the `attn` and `ln_2` sub-layers from each attention layer.

We downloaded pre-trained CLIP models from https://github.com/openai/CLIP, PyTorch ImageNet classifiers from https://pytorch.org/vision/stable/models.html and ConvNext models from https://huggingface.co/models?sort=downloads&search=facebook%2Fconvnext.

We trained the $\beta$-VAE model from scratch on the Places-365 dataset (https://pytorch.org/vision/main/generated/torchvision.datasets.Places365.html) with $\beta = 0.01$, Adam optimizer with default parameters, and a learning rate of 0.0001, for 13 epochs at a batch size of 64. The encoder consisted of five standard convolutional layers (with filter counts of 64, 265, 512, 512, and 512, respectively). The latent layer had a dimension of 2048. The decoder consisted of five standard de-convolutional layers (with filter counts of 512, 256, 256, 256, and 3, respectively). The kernel size for all layers was 5.

For further details, please see our code repository at https://github.com/c-j-bates/scaling-models-of-vwm-to-natural-images/tree/main. Our experiments were not preregistered.

**Reporting summary**. Further information on research design is available in the Nature Portfolio Reporting Summary linked to this article.

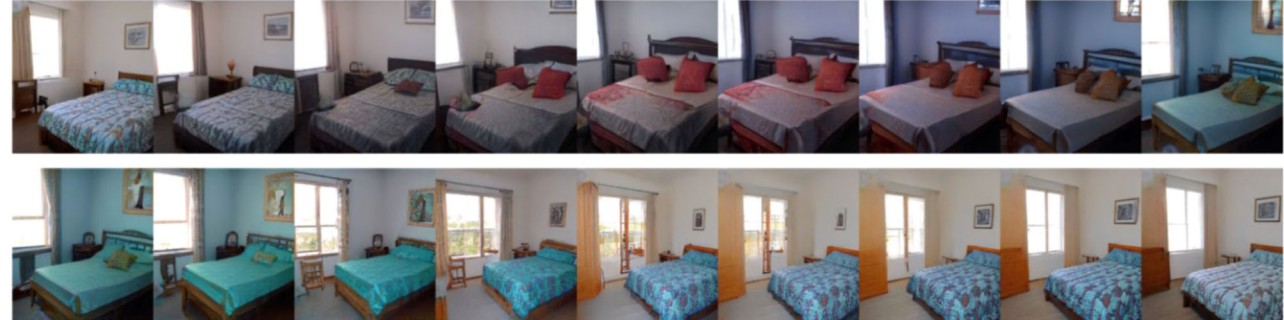

**Fig. 1 Evenly-spaced samples from one wheel in the Scene Wheels experiment (radius = 8).** A participant in a trial moved the mouse around a response wheel in order to recover the image that best matched their memory for the stimulus. The image displayed inside the response wheel smoothly morphed with an angular position along the wheel. Scanning from left to right (starting on the top row and continuing to the bottom row), one can observe how the image changes when moving along one such wheel.

## Results

**Continuous report with natural images**. To study VWM for natural images, we analyzed data collected by Son et al.[13]. Stimuli were generated using StyleGAN[14] (a generative adversarial network) trained to produce novel, naturalistic indoor scenes (Fig. 1). We will refer to this as the "Scene Wheels dataset". On each trial, participants performed a continuous report task. Visually, participants first saw a GAN-generated indoor scene, which subsequently disappeared for a short delay. On the response screen, they initially saw a wheel with an image in the center. As they moved the mouse around the wheel, the image gradually morphed, and their task was to locate the original image on the wheel. The image morphs corresponded to evenly spaced samples along the circumference of a circle drawn in the GAN's high-dimensional latent space. Specifically, each circle was drawn around a center-point in a randomly sampled 2D plane in the latent space. Trial difficulty was controlled at a coarse level by changing the radius of the circle. Larger radii resulted in more distinct response alternatives since they were further away from each other in code-space. The dataset includes 25 total "wheels" (circles in latent space), with five unique center points (each center belonging to a different random 2D plane) and five different radii around each center point.

*Model zoo.* We compare TCC models constructed based on a wide range of feature spaces, including layers from deep vision models and simpler baseline models. Our two simplest baselines are the raw pixel vectors (length $3 \times 256 \times 256$) and the RGB channel averages (length 3). We also include the latent representation from a $\beta$-Variational Autoencoder ($\beta$-VAE)[15] as a more sophisticated baseline. Deep autoencoder models have been explored as tools to learn better image and video compression algorithms for technological applications[16,17], as well as to model human visual memory[18–20]. Here, we consider it a baseline model because it is a much smaller network than our non-baselines. In addition to baseline models, we consider networks trained on the ILSVRC ImageNet classification challenge (both the 1000-way and 22,000-way versions) and networks trained on the contrastive language-image pre-training (CLIP) objective[21]. The CLIP objective is conceptually related to classification, but it encourages networks to learn semantically richer outputs that capture all the information contained in a typical image caption rather than a single class label. We selected a subset of pre-trained models provided by OpenAI, including models based on the ResNet-50 backbone (and larger variants of the same architecture), which is a convolutional network, and Vision Transformer, which is non-convolutional but also shown to be human-like[22]. For the ImageNet classifiers, we took several classic, pre-trained networks

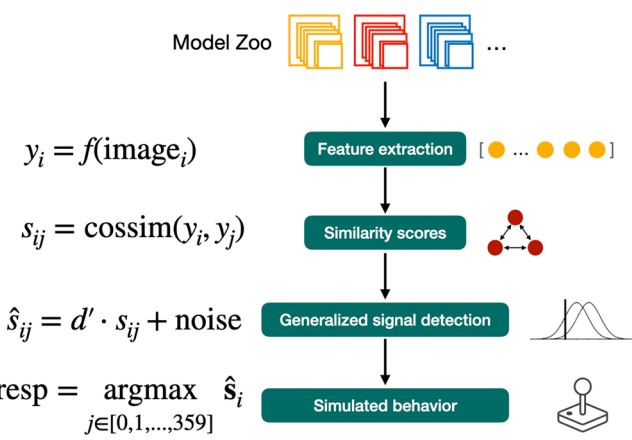

Model Zoo

$$y_i = f(\text{image}_i)$$

Feature extraction

$$s_{ij} = \text{cossim}(y_i, y_j)$$

Similarity scores

$$\hat{s}_{ij} = d' \cdot s_{ij} + \text{noise}$$

Generalized signal detection

$$\text{resp} = \underset{j \in [0,1,\dots,359]}{\arg\max} \hat{\mathbf{s}}_i$$

Simulated behavior

**Fig. 2 Schematic overview of our modeling pipeline using DNN features and the TCC model.** For a given DNN model and layer within that model, we take the (flattened) activations from that layer after feeding in a stimulus image and each response alternative from the scene wheel, in turn. There were always 360 evenly-spaced response options. For each option $j$, we computed the cosine similarity between that option's activation vector ($y_j$) and the stimulus's ($y_i$). After scaling by a constant factor $d'$, we added independent Gaussian noise with unit variance to each of the 360 similarity scores to produce corrupted similarity scores. Finally, we assumed responses were the argmax of these noisy scores.

from the Torchvision repository. We also took pre-trained ConvNext models[23] (a recent convolutional competitor to Vision Transformers) from Facebook's Huggingface repository. Finally, we took a "harmonized" version of ResNet-50 from the repository provided by Fel et al.[24], which is optimized to encourage classification decisions to depend on the same areas in the image that humans rely on when making the same decisions.

*TCC model.* We construct a separate TCC model for each layer in each architecture, as well as each baseline (see Fig. 2 for a schematic). For each trial, we compute all pairwise similarities between the target stimulus and each of the 360 options along the response wheel. We then multiply these 360 similarity scores by a scaler, $d'$, which corresponds to the memory strength for an exact match (similarity = 1), and therefore controls response accuracy. Finally, we add independent Gaussian noise with unit variance to each of the scores and take the option with the max score as the model's response on that simulated trial. (Note that it would be mathematically equivalent to scale the variance of the noise, rather than the similarity scores.) We simulated 8000 model responses for each trial in the dataset.

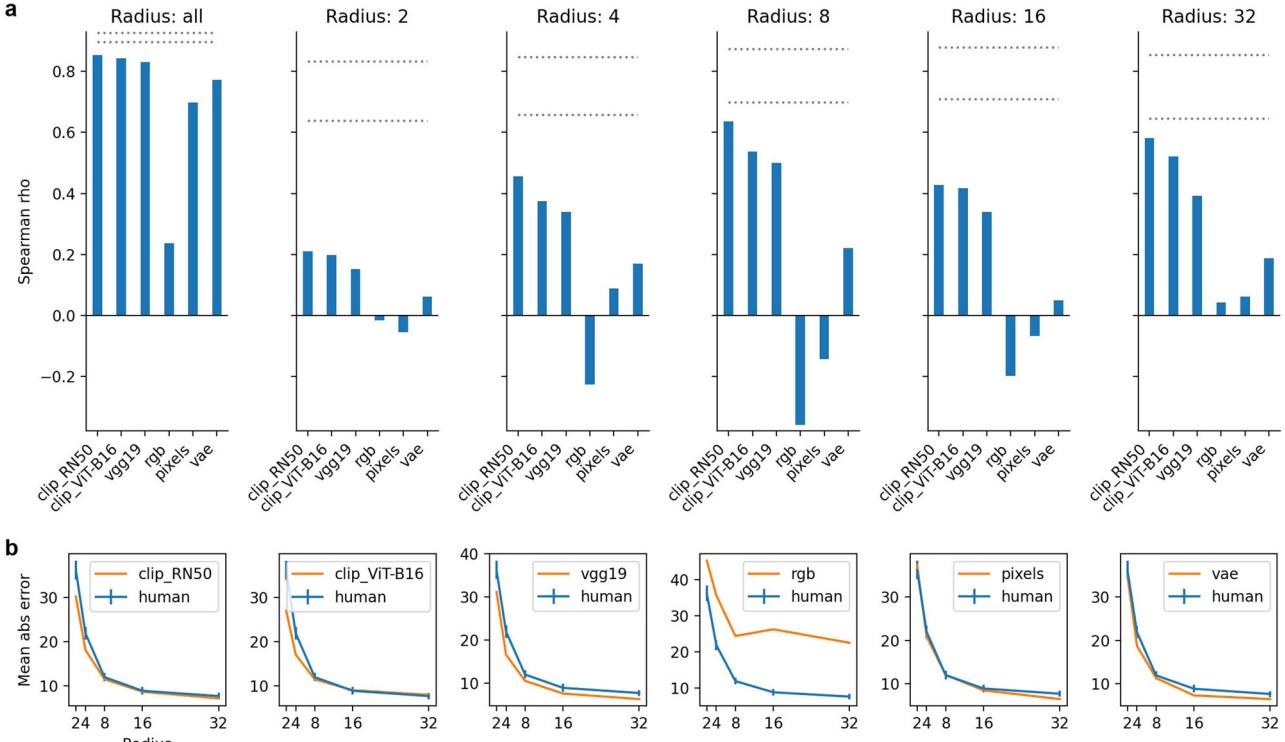

**Fig. 3 Rank-order difficulty results for Scene Wheels dataset ($N = 20$ participants). a** Spearman ranks correlations for trial difficulty between the best layer in selected DNN architectures, baselines, and humans. Blue bars indicate $p < 0.05$. Dotted lines are an indication of the noise ceiling. Specifically, we took bootstrap resamples of human responses within each radius, and for each resample we computed the Spearman correlation coefficient between it and the original data. The lines are the fifth and 95th percentile. **b** Comparison of human and model mean errors within each wheel radius. Error bars are bootstrapped with 90% confidence intervals.

*Trial difficulty rank-order analysis.* For each architecture considered, we searched for the layer that best matched human data. For each layer, we fit our only model parameter, $d'$, according to model likelihood. We conducted a grid search over $d'$ values and used a histogram approximation to the model likelihoods. We then estimated the Spearman correlation coefficient between the human and model mean absolute error per trial. Because there was a large number of unique stimuli compared to the number of responses collected, it was necessary to bin trials. (Note that nearby stimuli on a given response wheel tended to be highly similar.) We divided each scene wheel into 12 evenly sized bins, and for each bin, we averaged errors across all trials for which the target stimulus fell within that bin.

Results of the Spearman analysis are presented in Fig. 3a. When trials from all radii are aggregated, features taken from our selected CLIP and ImageNet classifier models capture trial-by-trial difficulty better than baselines. However, given the stimulus design for this experiment, the crucial test lies in how much variance can be explained within each radius. Because wheel radius modulates trial difficulty at a coarse level, even a relatively poor model can explain a fair amount of variance when aggregating trials across radii. When restricting our analysis to particular radii, our best models still beat out the baseline models, explaining some fine-grained variance in the rank-order of difficulty. As expected, baselines also had lower likelihoods (Table 1). As another way to compare models, we also plotted mean error per radius for humans and models (Fig. 3b). Consistent with the correlation results, both our VAE model and raw pixels capture the relationship between error and radius as well as our best models. The best-fitting (zero-indexed) layers tended to be past the midpoint of the architectures, specifically,

**Table 1 Comparison of models and baselines on Scene Wheels dataset.**

| Model | Log-likelihood | Spearman |
|---|---|---|
| RGB channel means | −25,722 | 0.24 ($p < 0.001$) |
| Pixels | −23,077 | 0.70 ($p < 0.001$) |
| VAE | −22,935 | 0.77 ($p < 0.001$) |
| CLIP RN50 | −22,628 | 0.85 ($p < 0.001$) |
| CLIP ViT-B16 | −22,647 | 0.84 ($p < 0.001$) |
| VGG-19 | −22,606 | 0.83 ($p < 0.001$) |

30 (of 36) for VGG-19, 24 (of 26) for CLIP ResNet-50, and 12 (of 23) for CLIP ViT-B16. Finally, see Supplementary for error density (Fig. S1) and scatter plots (Fig. S2), as well as Spearman results for all layers (Fig. S3).

We also conducted a comparison across DNN architectures to examine what factors might lead an architecture to explain more variance in this experiment (Fig. 4). We considered several dimensions, including number of images seen during training, type of architecture, and number of trainable parameters. Since we are unable to do an exhaustive search over these factors (and various confounds may exist), we present qualitative results, which may be suggestive for future work.

Overall, we find that architecture, number of trainable parameters, and number of training images may all be important factors. For each architecture, we selected the best layer according to its model likelihood. We find that the highest correlation is achieved by a CLIP pre-trained network, which also saw the most images during training, although some networks trained on the original ImageNet 1000-way classification dataset are competitive

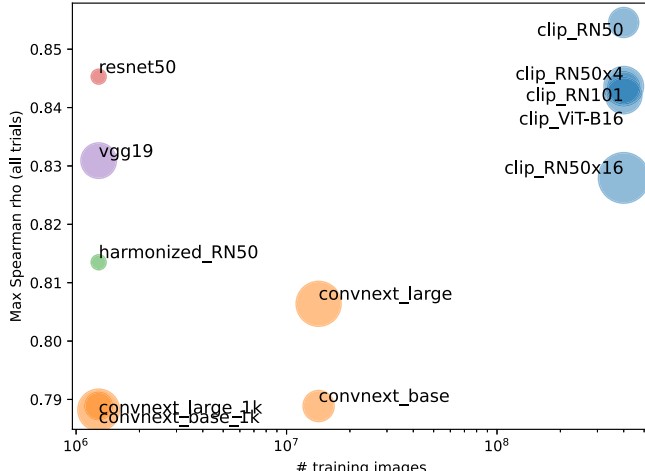

**Fig. 4 Extended cross-model comparison on Scene Wheels dataset.**
Marker radius is proportional to a number of trainable parameters.
ConvNext models labeled `1k' were trained on the 1000-way ImageNet
classification dataset and the others were trained on the 22,000-way
version.

with it. At the same time, we see that within the class of
ConvNext models, increasing the number of training images
increases correlation. However, the number of training images
may be confounded with the objective since the better-
performing ConvNext models were trained on the 22,000-way
classification task as opposed to the 1000-way. (Another
possibility is that training objectives that encourage richer
semantic information at the output layer lead to higher
correlations.) Keeping objective and training set fixed, we also
see that some architectures outperform others. Within CLIP-
trained models, the Vision Transformer does worse than several
convolutional architectures. Within models trained on ImageNet
1000-way classification, VGG-19 and ResNet50 outperform
ConvNext.

**Continuous report with color and orientation.** In our experi-
ments with artificial images, we analyzed previously collected data
from experiments studying color[25] and orientation[6] working
memory. Both experiments we analyzed used continuous report
tasks. In the color memory experiment, every item in each display
was probed, in turn. In the orientation experiment, one item was
probed at random. In both experiments, on each trial, partici-
pants first briefly viewed a display with a collection of items,
followed by a retention period, and finally a response screen.
Similar to the Scene Wheels experiment, participants moved the
mouse to select a point on the response wheel that best matched
their memory of the target item. The color dataset only includes
trials with set-size of three, and the stimuli were colored circles.
The orientation dataset includes set-sizes 1, 2, 4, and 8, and items
were oriented with colored lines.

In addition to examining rank-order of trial difficulty as above,
we aimed to explain set-size effects, as well as a subset of well-
known response biases and inter-item effects. We restricted our
evaluation to the same subset of well-performing models
presented in the Scene Wheels experiment. In each experiment,
we showed the same stimuli to our models as were shown to the
participants. For the response options, the unprobed items were
left intact, and the target item was varied.

Figure 5 shows the results of the rank-order difficulty analysis,
after refitting our selected DNN architectures to the color and
orientation datasets, separately. (Figs. S4 and S5 show the
unabridged results for all layers.) Based on these correlations

alone, it is unclear which model provides the strongest account
of orientation and color memory. Crucially, note that here
we have used absolute error rather than signed error in order to
be consistent with the previous analyses. As a consequence, these
results provide incomplete information about model fit. As we
visualize below, both human and model response biases vary
roughly as sinusoidal functions. This means that a model can be
mirrored along the vertical axis and yet result in a high
correlation under our analysis. In fact, CLIP ViT-B16 exhibits
this behavior (see Fig. 6, left). But even when the model and
human bias curves are mostly aligned in phase and frequency,
small misalignments can heavily penalize correlation values. Our
subsequent analysis addresses these shortcomings.

We next asked whether our models could explain response
inhomogeneities in color and orientation working memory. A
striking finding from orientation memory experiments is that
recall for nearly horizontal and vertical orientations is exagger-
ated away from these cardinal orientations (repulsion). At the
same time, responses are biased toward the oblique orientations
(attraction)[26]. In color working memory, there exists a set of
"focal" colors that responses are biased toward[27].

In both color and orientation, we find a qualitative correspon-
dence between the shape of the human bias function and at least a
subset of the models (Fig. 6). In the case of orientation, we find
the closest correspondence between human data and the VGG-19
model. To quantify this, we fit sine waves to human data and each
of the models (Table 2). While the amplitude is larger for the
human data than any of the models, both the phase and
frequencies are closely matched for both VGG-19 and CLIP
ResNet-50. Interestingly, the CLIP ViT-B16 model exhibits an
attraction bias toward the cardinal orientations, rather than
repulsion, suggesting convolutional architectures may contain a
more human-like representational bias than vision transformers
in this particular stimulus domain.

Also note that we produced the results in the left panels of
Fig. 6 by fitting to set-size 1, alone, whereas other reported results
on orientation relied on fits to all set-sizes simultaneously. The
reason for this choice was that human responses become noisier
with larger set-sizes (see noise ceilings in Fig. 5), and the strong
repulsion bias seen at set-size 1 gets washed out. By fitting set-size
1 separately, we can thus get a clearer sense of a DNN's ability to
explain this bias. However, see Supplementary Fig. S6 for results
when using a model fit to all set-sizes simultaneously. We found
that the results changed only for CLIP ResNet-50.

We next examined set-size effects in the orientation working
memory dataset, which included four set sizes (1, 2, 4, and 8). We
compared the mean absolute error per set-size between humans
and our models and found all best-fit models to exhibit a set-size
effect (Fig. 7a). We further investigated what causes mean error in
our models to vary as a function of set-size. We find that the
effect is caused by the sparsity of activations in the DNN layers.
As more objects are added to the background of the image, more
activations become non-zero, causing the range of similarity
values in the response options to shrink (Fig. 7b). Since noise with
fixed variance is added to these values, responses become more
easily corrupted with increasing set-sizes. This explanation bears
some resemblance to neural resource models of working
memory[8], which appeal to divisive normalization between
neurons in a population as the mechanism to control neural
resource allocation across items in the display. However, these
neural population models differ in that they predict a relatively
constant overall level of activation, whereas the DNN models we
examined increase their activation with set-size.

Next, we considered inter-item effects[5]. Found strong evidence
that memory errors for one item in a display depend on the other
items they appeared with. A specific hypothesis they tested was

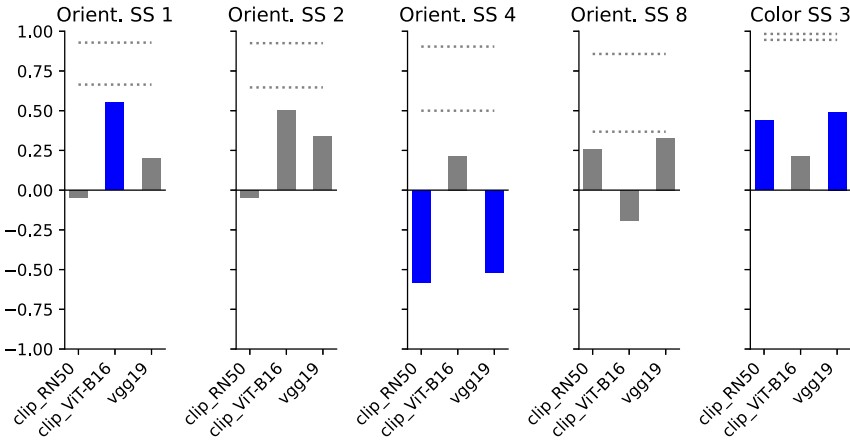

**Fig. 5 Results of the Spearman correlation analysis after refitting our models to color and orientation stimuli.** Blue bars indicate $p < 0.05$.

**a**
clip_RN50

clip_ViT-B16

vgg19

human

**b**
clip_RN50

clip_ViT-B16

vgg19

human

**Fig. 6 Comparison of humans' and models' average response bias in orientation and color memory tasks. a** Orientation experiment (set-size 1). **b** Color experiment (set-size 3). Red solid lines are sine wave fits to the data.

that people store hierarchical representations of the displays. At the upper level, they may record the overall level of dissimilarity between the items, while at a lower level, they record item-specific details. To test this hypothesis, they computed a correlation coefficient between two measurements. Specifically, for each trial, the two measurements were: (1) the circular variance of the three hues in the target display and (2) the circular variance of the three hues chosen by the participant. Intuitively, the correlation coefficient summarizes how homogeneity of hues within a display relates to the homogeneity of the responses. Importantly, the analysis only included trials for which participants were far off on all their responses (>45°). They found a significant correlation of 0.4. When we conducted the same analysis with our selected models, we found an insignificant correlation near zero for all of them, suggesting that our models do not capture this aspect of human behavior.

Finally, we asked whether the same or similar layers within each model provided the best explanations across experiments. For VGG-19, the best-fit layer for the Scene Wheels dataset was layer 30, but for color and orientation, it was 7 and 19, respectively. For CLIP ResNet-50, the best layer was 24 for Scene Wheels, 11 for color and 16 for orientation. For CLIP ViT-B16, the best layer was 12 for Scene Wheels, 23 for color, and 11 for orientation. (When fitting to orientation set-size 1 alone, the best-fit layer for CLIP ResNet-50 changed to 15.) Thus, for both convolutional architectures, the best layer was deeper for natural images than both the artificial experiments, but for the vision transformer-based architecture, this was not the case.

## Discussion

In this work, we combined several recent advances from cognitive science and AI to build *scalable* models of visual memory. We sought to build models that are not restricted to tasks with low-dimensional stimuli and/or simple feature reports but can make more general predictions. In particular, we sought to understand what features are stored in memory over the short term after viewing natural images. We then asked whether similar features are stored when viewing the kinds of sparse, artificial displays typically used in working memory experiments. We constrained our search for human-like features to two classes of pre-trained DNN, ImageNet classifiers and CLIP models.

In a continuous report task with GAN-generated, naturalistic indoor scenes, we found that our best models were able to capture people's memory errors better than several reasonable baselines. Surprisingly, layers from the same DNN architectures were also able to reproduce some important findings in the artificial stimulus domains of orientation and color, namely set-size effects, the repulsion bias in orientation memory, and the focal color bias in color memory. By contrast to prominent models of VWM, our TCC-based models explain responses purely on the basis of psychological similarity and representational geometry, and do not appeal to notions of information load. For example, set-size effects in multi-item displays are usually explained in terms of a limited resource (specified as bits, slots, or spike counts) allocated across items, and do not appeal to any notion of representational geometry. At the same time, these distinct classes of explanation may also be compatible. For instance, regularization schemes applied to DNN activations (e.g. L2 norm) can be seen as imposing a resource constraint, but they also change the representational geometry in critical ways. Possible equivalences between load- and similarity-based explanations open interesting avenues for future research.

Our models were built on the hypothesis that the features stored in VWM are noisy or compressed versions of features computed when initially perceiving a stimulus. However, this hypothesis could be tested in a more direct way by using neural recordings. Previous work has shown that when DNNs are trained to predict neural activity directly (e.g., using fMRI data), the learned representations recapitulate key behaviors and capabilities of human vision. For instance, when trained on activity from face-selective areas, the resulting representations are able to solve non-trivial segmentation problems, picking out faces in complex scenes[28,29]. The outputs of these networks, or even the fMRI data used to train them, could be directly swapped in for the features we used in the present work.

Finally, our approach may prove useful in clarifying some longstanding debates about the nature of VWM. In particular, our method allows us to ask how biases and capacity limits in certain artificial paradigms fit into a larger picture that includes behavior in more natural settings and tasks. Given that many important findings about VWM come from unnatural stimuli, an important baseline to test is whether adaptation to the demands of our natural environment explains these phenomena. Our results shed light on this and related questions. We found that a TCC model using the right DNN features could explain both set-size effects and response biases in color and orientation memory, despite only being trained to classify natural images. More work is

---

**Table 2 Parameters recovered from least-squares fit of sine wave to both human and model orientation response-bias data in Fig. 6.**

| Model | Amplitude | Phase (deg) | Frequency |
|---|---|---|---|
| CLIP RN50 | 0.22 | 6.53 | 4.0 |
| CLIP ViT-B16 | −2.58 | 5.38 | 4.0 |
| VGG-19 | 0.83 | −13.15 | 4.0 |
| Human | 4.01 | −0.22 | 4.0 |

Specifically, we fit the function $y = A\sin(\theta x + b)$, where $A, \theta, b$ are amplitude, frequency, and phase, respectively. Note that because the stimulus space is circular, repeating every $\pi$ radians, the frequency must be a multiple of 2. Fits were estimated using the `curve_fit` function from the SciPy Python package.

---

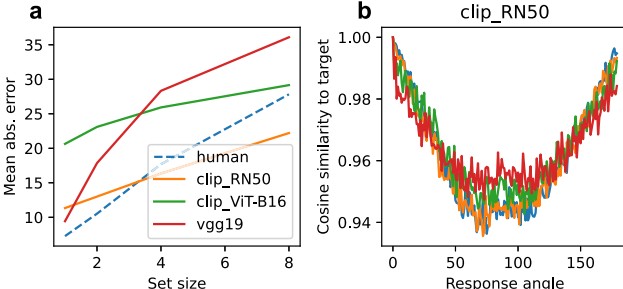
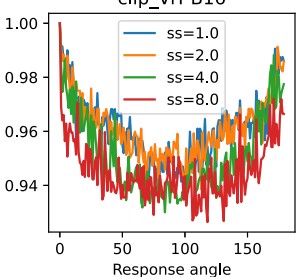
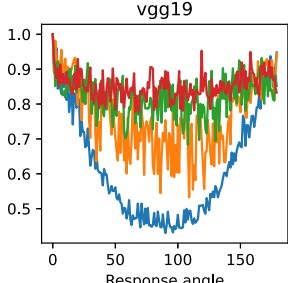

**Fig. 7 Analysis of set-size effects in TCC models. a** Comparison of human and model mean errors within each set-size in the orientation memory task. **b** Raw similarity scores (per set-size) for our three selected models between a target with a horizontal orientation and all response options.

necessary to determine whether these correspondences are simply coincidental or provide a satisfying explanation of human behavior. Nonetheless, our work constitutes a necessary first step toward more flexible and general models of visual memory that can accommodate findings from both natural and artificial stimulus domains.

## Data availability

We provide a repository that includes copies of the human data in all experiments. It can be found at https://github.com/c-j-bates/scaling-models-of-vwm-to-natural-images/tree/main. Data for the Scene Wheels experiment were downloaded from https://osf.io/h5wpk/[30]. Data for the orientation memory experiment were downloaded from https://osf.io/s7dhn/[31]. Data for the color memory experiment were obtained directly from the authors.

## Code availability

Code to reproduce all analyses can be found in our repository: https://github.com/c-j-bates/scaling-models-of-vwm-to-natural-images/tree/main. The DOI for the audited release is https://zenodo.org/records/10223343.

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

## Acknowledgements

This work was supported by the Multi-University Research Initiative Grant (ONR/DoD N00014-17-1-2961). The funders had no role in study design, data collection, and analysis, the decision to publish, or the preparation of the manuscript.

## Author contributions

C.J.B. and S.J.G. conceived of and developed the initial idea. C.J.B. conducted all analyses. S.J.G. and G.A.A. advised on the project and paper edits.

## Competing interests

The authors declare no competing interests.
