## [Peer Review File · Communications Psychology]

5th Oct 23

Dear Dr Bates,

Thank you for your patience during the peer-review process. I am sorry that it took us longer than anticipated to return to you with a decision.

Your manuscript titled "Scaling models of visual working memory to natural images" has now been seen by 3 reviewers; this includes two referees who vetted the work at the original journal (Reviewers #2 and #3), and one additional reviewer (Reviewer #4) whom we recruited to undertake code review. I include their comments at the end of this message.

As you will see, the original reviewers are enthusiastic about the work and the revisions you have undertaken and have no remaining concerns. The referee who vetted the code and data underlying your work raised some issues, primarily relating to documentation and accessibility.

We are very interested in the possibility of publishing your study in Communications Psychology, but would like you to address these concerns before we make a decision. We therefore invite you to revise and resubmit your manuscript, along with a point-by-point response to the reviewers.

Please note that your revised manuscript must comply with our formatting and reporting requirements, which are summarized on the following checklist: <https://www.nature.com/documents/commspsychol-style-formatting-checklist-article-rr.pdf> Communications Psychology formatting checklist and also in our style and formatting guide <https://www.nature.com/documents/commspsychol-style-formatting-guide-accept.pdf> Communications Psychology formatting guide .

Please use the following link to submit your revised manuscript, point-by-point response to the referees' comments (which should be in a separate document to any cover letter) and the completed checklist:
[link redacted]

Please do not hesitate to contact me if you have any questions or would like to discuss these revisions further. We look forward to seeing the revised manuscript and thank you for the opportunity to review your work.

Best wishes,

Marike

Marike Schiffer, PhD
Chief Editor
Communications Psychology

EDITORIAL POLICIES AND FORMATTING

Editorial Policy: <https://www.nature.com/documents/nr-editorial-policy-checklist.pdf> Policy requirements (Download the link to your computer as a PDF.)

* **CODE AVAILABILITY:** All Communications Psychology manuscripts must include a section titled "Code Availability" at the end of the methods section. In the event of publication, we require that the custom analysis code supporting your conclusions is made available in a publicly accessible repository; at publication, we ask you to choose a repository that provides a DOI for the code; the link to the repository and the DOI will need to be included in the Code Availability statement. Publication as Supplementary Information will not suffice. We ask you to prepare code at this stage, to avoid delays later on in the process.

* **DATA AVAILABILITY:**

All Communications Psychology manuscripts must include a section titled "Data Availability" at the end of the Methods section or main text (if no Methods). More information on this policy, is available at <http://www.nature.com/authors/policies/data/data-availability-statements-data-citations.pdf>.

At a minimum the Data availability statement must explain how the data can be obtained and whether there are any restrictions on data sharing. Communications Psychology strongly endorses

open sharing of data. If you do make your data openly available, please include in the statement:

We recommend submitting the data to discipline-specific, community-recognized repositories, where possible and a list of recommended repositories is provided at <http://www.nature.com/sdata/policies/repositories>.

If a community resource is unavailable, data can be submitted to generalist repositories such as [figshare](https://figshare.com/) or [Dryad Digital Repository](http://datadryad.org/). Please provide a unique identifier for the data (for example a DOI or a permanent URL) in the data availability statement, if possible. If the repository does not provide identifiers, we encourage authors to supply the search terms that will return the data. For data that have been obtained from publicly available sources, please provide a URL and the specific data product name in the data availability statement. Data with a DOI should be further cited in the methods reference section.

REVIEWERS' COMMENTS:

Reviewer #2 (Remarks to the Author):

I have reviewed a previous version of this manuscript for a different journal. The authors have done a good job addressing my previous concerns in the revised manuscript. I have no further concerns.

Reviewer #3 (Remarks to the Author):

The authors have sufficiently addressed my concerns (raised in my previous review for Nat Comms.) and made suitable changes to the manuscripts. The modifications to the results further support the claims being made. As such I do not have any more suggestions for changes and consider the manuscript of good quality scientifically for publication.

Reviewer #4 (Remarks to the Author):

Overall, I couldn't find any significant issues in the code. The code also aligned with the paper. I found no sign of tempering in the modeling and figure-drawing process. So I have no problem with the paper.

However, I have some issues with the data and documents available. Following the link in the paper, while I can find all the code needed to create the analysis in the paper, I failed to find all the data except the data from Bayes 2014 paper. Not having data available means that I can not verify the accuracy of the results in the paper, as the tempering could happen in the raw data. All the code checks out, so I don't believe that the authors tempered the data. I just couldn't be 100% sure of the validity. Lacking the data also makes verifying the code very difficult.

Another minor issue is the documentation. The authors listed the packages used in the code, which is always great. However, I still had to install several packages not listed in the documents. So it might be worth going through the dependency again with a clear python environment. The documentation did not list the entire code usage but provided a few examples. While it's accessible through the code, it would be very grateful for the authors to list all the usages.

Since I was asked to review code instead of reviewing the paper. So that's most of my review. However, as someone who does similar research. I do find that the continuous scene stimuli very interesting. I would really love to see the author also release the documents regarding how the stimuli were generated. It opens another way to approach the continuous materials, which can be applied beyond working memory studies. I do understand that documentation takes effort, so it's up to the authors.

Below are our replies to R4.

Overall, I couldn't find any significant issues in the code. The code also aligned with the paper. I found no sign of tempering in the modeling and figure-drawing process. So I have no problem with the paper.

However, I have some issues with the data and documents available. Following the link in the paper, while I can find all the code needed to create the analysis in the paper, I failed to find all the data except the data from Bayes 2014 paper. Not having data available means that I can not verify the accuracy of the results in the paper, as the tempering could happen in the raw data. All the code checks out, so I don't believe that the authors tempered the data. I just couldn't be 100% sure of the validity. Lacking the data also makes verifying the code very difficult.

The original data for the color experiments can be found at brady_alvarez/dataTestAllDegreesOrd.mat. This file is loaded in the `TCC_modeling.py` script in the `'load_brady_alvarez'` function.

The original data for the scene wheels experiments can be found at scene_wheels_mack_lab_osf/Data/sceneWheel_main_data_n20.csv. This file is loaded in the `TCC_modeling.py` script in the `__init__` of `TCCSceneWheel`. We have added clarification for this to the README.

Another minor issue is the documentation. The authors listed the packages used in the code, which is always great. However, I still had to install several packages not listed in the documents. So it might be worth going through the dependency again with a clear python environment. The documentation did not list the entire code usage but provided a few examples. While it's accessible through the code, it would be very grateful for the authors to list all the usages.

We've done as instructed and added packages found to be missing from the original list. As for usage, we've provided the key commands to reproduce the paper results.

Since I was asked to review code instead of reviewing the paper. So that's most of my review. However, as someone who does similar research. I do find that the continuous scene stimuli very interesting. I would really love to see the author also release the documents regarding how the stimuli were generated. It opens another way to approach the continuous materials, which can be applied beyond working memory studies. I do understand that documentation takes effort, so it's up to the authors.

The methodological details for the GAN-generated scene wheels can be found in the original methods paper referenced in our work, as well as their github repo.

PREVIOUS ROUND OF REPLIES

To all reviewers:

We have made significant changes to the results for the color and orientation experiments. First, and most importantly, we discovered an error in the color WM dataset (response-error data field had a flipped sign!). The result of fixing that error is that the human response-bias plot for color WM is flipped vertically from before (Fig. 7, bottom right), and we see now that the correspondence between models and humans is quite strong, in fact.

Second, we have updated our interpretation of the spearman correlation results (Fig. 6). We have added a discussion of the limitations of that analysis in evaluating our models in this domain. Specifically, we recognized that the periodic and sinusoidal nature of the data makes interpretation of the correlations fraught. We have fit sine waves to the bias curves to emphasize the match between models and humans in terms of the general shape of the waves (i.e. phase and frequency).

With these two updates, we find our model can reasonably accommodate both naturalistic and artificial stimuli.

Note that we have also made a small modification to Fig. 4 in order to be more consistent with the other analyses, and removed several analyses that we now find to be unhelpful to the reader.

Reviewer #1

This paper combines three modern tools to understand visual working memory: continuously varying, artificially generated, but naturalistic images (“Scene Wheels”), the powerful and flexible TCC model of working memory, and a wide variety of front-end DNN architectures for feature extraction. The DNN+TCC models are compared to human data using mean absolute error in the Scene Wheels data. Then, the same models are applied to one-dimensional stimuli typically used in the visual working memory literature.

The paper addresses an important and understudied question in the field. Most researchers of visual working memory are hesitant to move beyond the safe confines of extremely simple stimuli, so every serious attempt to do so must be embraced. The paper’s combination of tools is innovative, effective, and methodologically rigorous. I really appreciate the authors’ brutal honesty in assessing their models’ failure to account for some effects found with simple stimuli. It would have been easy to leave this test out altogether or to only report the parts that work, but that would have been a disservice to the field. Overall, this paper is a valuable contribution to the field.

I have two questions:

- The paper only uses mean absolute error as a metric. Since much of the VWM literature has revolved around the shape of the error distribution (whether stimulus-dependent or not), it seems useful to assess the networks also on the full error distribution.

We have now added error density plots to the Supplementary.

- Did the authors try a more biologically inspired front end, such as the steerable pyramid?

This is an interesting suggestion. However, changing the front end to the model would require retraining all of the networks, which are optimized for standard RGB values. This would require substantial extra work, since retraining SoTA deep architectures from scratch can be nontrivial. More importantly, it's our impression of the visual neuroscience literature that these sorts of front ends have been largely superseded by networks trained end-to-end (see for example the work of Dan Yamins). These newer models still capture many aspects of early visual receptive fields, without manual construction. Thus, it's not clear to us that using classical front ends would add much.

Reviewer #2

This manuscript applied a recent cognitive model of visual working memory, the Target Confusability Competition (TCC) model, to the natural scene image domain. The original goal of the TCC model was to simulate human memory errors by scaling the psychological distances embedded in feature spaces, such as a color wheel, with the memory strength parameter, d' . This study focused on the psychological distance part of this model and questioned what features should be used to obtain psychological distances in order to correctly simulate the natural scene memory by TCC. The authors leveraged features extracted in individual layers of various DNNs as candidates for complex features represented by the human memory system. They obtained the psychological distances of those DNN-driven features and examined several aspects of working memory for natural scenes. Their main finding was that TCC fitted with DNN feature spaces predict human data better than the baseline feature spaces (pixels). However, those DNN features did not capture the trial-by-trial memory difficulty explained by image statistics. Also, those features could not simulate unique characteristics of the human memory system when applied to artificial lab experiment stimuli (e.g., an array with color patches).

I highly agree on the importance of this research question and I found the approach using DNN features for natural stimuli interesting and valid. However, I am not convinced that the finding significantly expands our knowledge of VWM in a naturalistic stimulus domain.

1. The layer selection was arbitrary (best match to human data), making the interpretations of the selected feature space confusing. When looking at Fig S1, many of the other layers that

were not selected were also significantly correlated with human memory performance, but with no reasonable patterns. In this sense, can we really conclude that the feature space in the selected layer meaningfully contributes to explaining the features used by the human memory system? I would ask the authors to provide a more careful interpretation of this point to make their findings more generally interpretable.

Interpretability is difficult with these networks: we lack strong principles with which to narrow the search *a priori*. However, it is reasonable to expect many layers to be at least somewhat human-like since they will usually carry *some* visual information. Our analysis necessarily has an exploratory quality to it, due to a lack of top-down guidance on the search over features. Discovering these kinds of principles should be a goal for future work. Nonetheless, there are some reasonable patterns. For example, deeper is generally better for CNNs with the scene wheel dataset. ViT is somewhat different, which is interesting but not shocking given how different the inductive biases are for transformers compared to convolutional networks. In general, it does not seem problematic to us that multiple layers from a network are highly correlated with human data, since adjacent layers in networks are highly correlated to begin with, and pairs of layers may capture both unique and overlapping aspects of human responses depending on precisely what transformation of the pixels they each compute.

2. In Fig 3, “Radius:all” panel shows that some baseline features were also quite highly correlated with human data, almost similar to the DNN features. Even pixels explained human VWM errors quite well, although we know that the visual system does not represent pixels per se as important features. Thus, I am unsure how valid the DNN features are as representations of the features used by the human visual memory system.

Please see our response to point 3. The key point is that the aggregated results have limited value due to the fact that they obscure the key within-radius variation. The fact that pixels could explain the aggregate results reasonably well reflects the fact that they can capture the general pattern that difficulty decreases with radius, but they do a poor job explaining within-radius variation.

3. Another somewhat confusing point is the discrepancy between this result and within-radius results. Within each radius, the baseline features showed low correlation with humans, but when pooling data across radii (left-most plot in Fig. 3), they were. What would bring this difference?

We have clarified our explanation of this phenomenon. To summarize, the experimental design involved a variable (the wheel radius, R), which modulated difficulty by making response options more or less distinct from each other. Thus, one needs to control for R in order to have a stronger test of the models. As a result, within-radius results are especially critical when comparing models. Stated differently, aggregating across R is a weaker test of a model.

4. Fig. 4 depicted very interesting qualitative data. But I think the interpretation of this result could be framed more related to the human memory system, that is, how do different aspects of DNNs related to characteristics of the human memory system. The current interpretation in the

manuscript (or at least, the way it's framed) is a lot more concerned with implications for DNNs than the human WM system.

Does the reviewer have any specific characteristics in mind? We would be open to suggestions. Given the looseness of the connection between CNN and Transformer architectures and human visual processing, we opted to avoid drawing mechanistic comparisons. For us, the geometry of the representation space for a chosen layer is key. From this standpoint, architectures that differ mechanistically from human vision may nonetheless produce representational geometries that match humans and still provide scientific insights. This can be true, for instance, if the more important factors in determining geometry are the training set and the objective function. Of course, architectural constraints may play a key role as well, and the best models may ultimately result from better matching the architecture of human vision. We see this as a good direction for future work, but outside the scope of this paper.

5. In Fig 5, the Trial-by-trial difficulty was not captured by DNN-based TCC models. This result says that the DNN features were not something human represents, casting doubt on the implication of this study. How can this apparent contradiction be reconciled?

We have now removed this analysis entirely, as it was needlessly confusing, and not all that informative. Instead, we have included a set of scatter plots to visualize each model's error compared to human error, within each scene wheel radius.

Minor:

6. The methods could be described more clearly for a broader readership. For example, more details for part 2 (application to color and orientation stimuli) would be helpful. At first, it was a little confusing if the psychological distances were calculated from the similarity between the displays (e.g., an array with multiple color/orientation patches) or a single color/orientation patch. I assume the former was the case, given the explanation for Fig. 7 results, but if this was the case, it was not easy to understand how to obtain the results in Figure 8, from the scaling based on stimulus display.

We have now added some text to speak to this detail. In brief, the entire stimulus image was always passed into the network, including all unprobed items.

7. Page 7: The authors describe an interesting observation that very large models perform worse at predicting human performance than more moderately sized models. This appears a specific case of the bias-variance trade-off that is commonly observed in machine learning. It might be worth mentioning this connection for the readers' benefit.

We have removed this section in the revised version. After redoing the DNN comparison plot to choose based on model likelihood rather than correlation, we found a less clear pattern in this respect and opted to withhold this speculation.

Reviewer #3

General summary:

The central demonstration of this paper is that higher pairwise similarity in the deep neural network (DNN) layer's representational space predict higher confusability in people's working memory (WM). The DNNs are better models of natural image WM as opposed to WM based on simpler stimuli (colours/orientations). The overall goal is build image-computable models of human visual working memory (VWM) performance on natural images, using DNNs. The authors have succeeded in this goal. However, it is unclear what new insights into the nature of VWM, conceptually or neurally, this work yields. The authors argue that the current paper is *the first* computational modelling project of natural image based VWM. As such, the well-established approach - comparing DNN activations to human behaviour - applied to VWM is a useful, but incremental, demonstration.

We believe this work is more than incremental. Our framework is the first to be able to generalize across all kinds of images. This generality forces a fundamental reexamination of previous theories of capacity limits and biases in VWM, and facilitates novel conceptual questions.

The conclusions and claims, for most part, do follow from the results - potential improvements are mentioned in the comments below. The methods used are well-established and the discussion following the observations is sufficient. The methods are mostly explained well - I have commented below where they could use improvement.

A suggestion: The addition of a set of scripts to reproduce the plots in the paper would be very useful for anyone in the field of VWM wanting to experiment with these models.

Please note that a script to produce figures already resides in our repo, in a file named `tcc_plotting.py`

Comments (in no particular order):

1. In the abstract you wrote, "people may rely on distinct cognitive systems or brain areas in artificial versus natural task domains". Presumably this is borne out of the observation that in CNNs the best layer predictive of human behaviour was deeper for natural images than for the artificial experiments. However, the correspondence between the depth in CNNs and depth in the ventral stream is straightforward (Sexton & Love, Sci. Adv. 2022; esp, if you factor in recurrent and top-down computations). Doesn't this claim require more evidence?

We agree with the reviewer that this was overly speculative. Our revised abstract no longer includes it.

2. I am not well-read into this field, so I cannot judge the novelty of this modelling approach on natural images. However, I do know that researchers have studied VWM for natural images/objects (Kaiser et al. Psych. Bull & Rev. 2015; Liu & Jiang JoV 2005; Jiang et al. JEP:HPP 2016; Tanabe-Ishibashi APP 2023; etc). It would be useful to discuss what insights this work adds to this field, beyond the estimation of VWM behaviour using a ANN instead of a-priori ratings of various stimulus attributes, etc. that other studies have employed.

While there have been previous studies of VWM for natural images, these studies have not developed a model that can explain the relevant data. Our key contribution is to build a bridge between the quantitative models of VWM for artificial images and the largely unquantified data on VWM for natural images. This is an important step towards establishing the generality of computational ideas about VWM.

3. In the introduction you wrote, “We argue that the most parsimonious hypothesis is that VWM is primarily built on top of feature detectors residing in the visual stream”. However, what the actual argument was to justify this statement was unclear. It might be useful to add references discussing the role of representations in visual cortex in VWM (in terms of driving PFC and other regions or the activity-silent WM literature) to clarify the argument.

We believe our argument is agnostic as to the particular role of the PFC in memory maintenance. Logically, the inputs to memory must be some function of representations in visual cortex. The more complex that linking function between perception and memory is, the more computationally or energetically costly. Thus, an organism should opt to reuse computations to the extent possible, and avoid complex (costly) transformations. We have modified the wording to emphasize the idea of computational reuse.

4. To understand the stimuli and the actual experiments which resulted in the behaviours being analysed, we have to read 3 more papers. As a reviewer, I can do that. As a reader I won't have time to do so and the reading would become much smoother if the methods details were shortly including in the figures i.e. in Fig. 1 & Fig. 6. Additionally, you could describe the methods in the captions so as to make “reading the paper through the images” possible.

We have now added additional details about these experiments to the text.

5. In terms of statistical analysis, it is unclear which of the differences between the results in the figures are reliable. As you have access to many participants (the N could also be mentioned in the captions), isn't it possible to show the confidence intervals across the pairwise model-behaviour correlations per participant? You could also plot the noise ceilings to quantify how robust across-participants the behaviour data is in the first place. This holds for all the results figures.

We previously included a noise-ceiling analysis in the top panel of Fig. 3, and have now added error bars to the lower panel of that plot, as well as noise ceilings to the analogous bar plots for color and orientation. We've also added the number of participants to the caption of Fig. 3.

6. In terms of making the figures more readable, might it be useful to draw a baseline (0) horizontal line in Figs. 3 & 6?

Done.

7. It is unclear why the beta-VAE is considered a baseline. Usually they are as expressive in terms of visual features as networks trained for object recognition. Is the particular beta-VAE in use trained, for e.g., on a simpler dataset, that makes it so different? In the results it also performs worse than the classification-trained CNNs. Why do you think that is the case?

beta-VAEs are trained on a pixel-loss objective which treats all pixel errors independently. Classification networks, on the other hand, are directly incentivized to encode images in ways that are aligned with common human goals (e.g. recognizing objects or affordances within a scene). In addition, our VAE had many fewer parameters and layers than the pretrained networks we used. It was trained on Places 365, which is comparable in size to ImageNet-1k. We have added these architectural details to the Methods.

8. In Fig.5. you showed that a linear combination of summary statistics explained some variance in human mean errors across stimuli. The models did not account for this variance. However, in the previous figures the models do explain variance in the human errors. One, is there an interpretation of what the linear combination entails? Are there certain statistics that do well than others to begin with? What do you think are the characteristics of this “independent” shared variance between human errors and models? This also feeds into the broader question of what the differences in explained variance between models and their layers actually reflect. This is partly dealt with in Fig.4. however it would be very informative to assess the “visual features” that account for the shared variance.

We have now removed this analysis entirely, as it was needlessly confusing and not all that informative. Instead, we have included a set of scatter plots to visualize each model's error compared to human error, within each scene wheel radius.

9. On pg. 10 you mention "they computed a correlation coefficient between circular variances, where one vector comprised the variances of the three hues in each stimulus display and the other was the variances of the three chosen hues at response time". What are these variances? Across what measure? This is unclear.

We have updated the language to (hopefully!) make this less confusing.

10. In terms of finetuning CLIP, did you train a new readout while keeping the rest of CLIP vision backbone frozen or did you fine-tune the entire RN50/ViT backbone?

We fine-tuned the whole network. However, in the revised version, this analysis has been removed, because we discovered that we can actually do a better job on artificial images with the non-fine-tuned networks than we had originally thought.

Nice work!

~ Sushrut Thorat

3rd Nov 23

Dear Dr Bates,

Your manuscript titled "Scaling models of visual working memory to natural images" has now been seen by our reviewer, whose comments appear below. In light of their advice I am delighted to say that we are happy, in principle, to publish a suitably revised version in Communications Psychology under the open access CC BY license (Creative Commons Attribution v4.0 International License).

We therefore invite you to revise your paper one last time to address a list of editorial requests. At the same time we ask that you edit your manuscript to comply with our format requirements and to maximise the accessibility and therefore the impact of your work.

Please note that it may still be possible for your paper to be published before the end of 2023, but in order to do this we will need you to address these points as quickly as possible so that we can move forward with your paper.

EDITORIAL REQUESTS:

SUBMISSION INFORMATION:

OPEN ACCESS:

Communications Psychology is a fully open access journal. Articles are made freely accessible on publication under a [CC BY license](http://creativecommons.org/licenses/by/4.0) (Creative Commons Attribution 4.0 International License). This license allows maximum dissemination and re-use of open access materials and is preferred by many research funding bodies.

For further information about article processing charges, open access funding, and advice and support from Nature Research, please visit <https://www.nature.com/commspsychol/article-processing-charges>

At acceptance, you will be provided with instructions for completing this CC BY license on behalf of all authors. This grants us the necessary permissions to publish your paper. Additionally, you will be

asked to declare that all required third party permissions have been obtained, and to provide billing information in order to pay the article-processing charge (APC).

* **DATA AVAILABILITY:**

[link redacted]

Best wishes,

Marike

Marike Schiffer, PhD
Chief Editor
Communications Psychology

REVIEWERS' COMMENTS:

Reviewer #4 (Remarks to the Author):

I'm happy with the changes made in the repo and the paper.